# Current Medical Therapy and Revascularization in Peripheral Artery Disease of the Lower Limbs: Impacts on Subclinical Chronic Inflammation

**DOI:** 10.3390/ijms242216099

**Published:** 2023-11-08

**Authors:** Andrea Leonardo Cecchini, Federico Biscetti, Matteo Manzato, Lorenzo Lo Sasso, Maria Margherita Rando, Maria Anna Nicolazzi, Enrica Rossini, Luis H. Eraso, Paul J. Dimuzio, Massimo Massetti, Antonio Gasbarrini, Andrea Flex

**Affiliations:** 1Cardiovascular Internal Medicine, Fondazione Policlinico Universitario A. Gemelli IRCCS, 00168 Rome, Italy; 2Facoltà di Medicina e Chirurgia, Università Cattolica del Sacro Cuore, 00168 Rome, Italy; 3Division of Vascular and Endovascular Surgery, Thomas Jefferson University, Philadelphia, PA 19107, USA; 4Department of Cardiovascular Sciences, Fondazione Policlinico Universitario A. Gemelli IRCCS, 00168 Rome, Italy; 5Department of Internal Medicine, Università Cattolica del Sacro Cuore, 00168 Rome, Italy

**Keywords:** peripheral artery disease, lower extremity arterial disease, residual risk, atherosclerosis, inflammation

## Abstract

Peripheral artery disease (PAD), coronary artery disease (CAD), and cerebrovascular disease (CeVD) are characterized by atherosclerosis and inflammation as their underlying mechanisms. This paper aims to conduct a literature review on pharmacotherapy for PAD, specifically focusing on how different drug classes target pro-inflammatory pathways. The goal is to enhance the choice of therapeutic plans by considering their impact on the chronic subclinical inflammation that is associated with PAD development and progression. We conducted a comprehensive review of currently published original articles, narratives, systematic reviews, and meta-analyses. The aim was to explore the relationship between PAD and inflammation and evaluate the influence of current pharmacological and nonpharmacological interventions on the underlying chronic subclinical inflammation. Our findings indicate that the existing treatments have added anti-inflammatory properties that can potentially delay or prevent PAD progression and improve outcomes, independent of their effects on traditional risk factors. Although inflammation-targeted therapy in PAD shows promising potential, its benefits have not been definitively proven yet. However, it is crucial not to overlook the pleiotropic properties of the currently available treatments, as they may provide valuable insights for therapeutic strategies. Further studies focusing on the anti-inflammatory and immunomodulatory effects of these treatments could enhance our understanding of the mechanisms contributing to the residual risk in PAD and pave the way for the development of novel therapies.

## 1. Introduction

Peripheral artery disease (PAD) represents one of the three main expressions of the atherosclerotic involvement of arterial beds, together with coronary artery disease (CAD) and cerebrovascular disease (CeVD). Over the past three decades, PAD has experienced a dramatic increase in prevalence and incidence worldwide [1]. The resulting increase in morbidity and mortality associated with PAD is further reflected in the economic burden imposed to society, as the annual expenditure per individual with PAD is three times higher when compared to people without PAD [2].

Despite significant improvements in the treatment of traditional cardiovascular risk factors, there is a substantial number of patients with PAD that experience significant progression of disease burden and poor outcomes [3,4,5]. This residual risk and disease progression suggest a plausible resistance to traditional therapies that have failed to address other pathophysiological disease pathways [6].

Inflammation, once thought to be a simple consequence of atherosclerosis, has progressively emerged as a solid pathogenic component that not only starts the development of atherosclerotic plaque but also accelerates its progression [7]. Traditional PAD therapies aim to restore blood flow, control cholesterol levels, lower blood pressure values, reduce serum glucose concentrations, and manage the prothrombotic burden. However, despite the extensively documented role of inflammation in PAD, the anti-inflammatory effects of current treatments and their impact on residual risk reduction have not been clearly established. 

A growing body of evidence supports the use of new immunomodulatory and anti-inflammatory drugs in atherosclerosis [6,8,9,10]. In this review, we aimed to provide an in-depth analysis of the available evidence of the role of traditional and novel therapeutic interventions in modulating atherosclerosis-associated inflammation, and their role in the treatment of symptomatic PAD.

## 2. PAD of Lower Limbs: The Alter Ego of CAD

The understanding and perception of heart disease by the general population are increasing through improvements in social awareness. On the contrary, PAD of the lower limbs remains mostly underdiagnosed, poorly understood, and is an often neglected medical condition [11]. It is estimated that over 230 million people worldwide suffer from PAD. Almost one out of two patients with CAD are concomitantly affected by an overt or non-overt condition of PAD [12,13].

Many factors contribute to PAD underestimation. Commonly, atherosclerosis is perceived as a disease localized primary in the coronary or cerebrovascular arteries. Moreover, reduced symptom awareness and/or the presence of confounding factors that either mimic the clinical expression of PAD or limit the physical activity of the patient cause a delay in the diagnosis and/or disease-specific intervention. This results in an extended natural progression of the disease burden that perhaps explains the high short-term mortality and morbidity rate after clinical diagnosis, which can be higher than common cancers (e.g., breast or prostate cancer) in selected populations [14].

Despite the disabling consequences associated with this condition, a diagnosis can be achieved rapidly and with non-invasive tests such as the ankle brachial index (ABI) and arterial echo color Doppler ultrasound [15]. An early detection opens up the potential for effective strategies to control the disease, slowing down the progression and reducing PAD-related morbidity and mortality, even before the need for a limb salvage revascularization procedure. 

In the majority of patients with PAD, a substantial change in lifestyle, the appropriate management of traditional cardiovascular atherosclerotic risk factors (i.e., blood pressure, taking control over diabetes, and reducing low-density lipoprotein cholesterol (LDL-C) serum levels), and abstinence from detrimental habits such as smoking result in significant decreases in morbidity and mortality [16]. Along with lifestyle corrections, pharmacological therapy is a mainstay for PAD management. More recently, for symptomatic patients with PAD, antithrombotic therapy may further reduce the mortality and disabling consequences resulting from cardiovascular events [17]. 

Unlike CAD, in which well-defined strategies are implemented in patients’ care, obtaining positive results, and despite the interventions described above, PAD frequently progresses to its most advanced stage of chronic limb-threatening ischemia (CLTI) due to suboptimal management, and sometimes even under appropriate therapeutic regimens [18]. CLTI often leads to the loss of limbs, a significantly lower quality of life, and even death. At this stage, pain and tissue loss are irreversible conditions that require timely invasive intervention to restore limb perfusion and prevent amputation. Nearly three out of four patients with CLTI undergo revascularization procedures [19].

So far, the management of PAD has been divided into three main phases, including the prevention of major risk factors with non-pharmacological interventions, the treatment of the diseases causing the progression of PAD with the use of the latest medications, and the relief of symptoms via the invasive restoration of blood flow in the ischemic limb. In addition to this, supervised exercise therapy (SET) is an incredibly simple yet innovative option to improve disabling ischemic claudication by stimulating the recruitment of collateral blood flow pathways, resulting in an improved quality of life and with substantial reductions in morbidity associated with revascularization, particularly in patients with mild or moderate obstructive disease. SET is also associated with an overall improvement in cardiovascular performance in patients with PAD. However, SET application at a population-wide level has significant limitations due to numerous social/environmental and practical barriers to establishing effective SET programs [20]. Non-pharmacological treatments, drugs, and invasive therapeutic procedures are aimed at the prevention, treatment, and symptomatic management of PAD resulting from obstructive arterial disease. The impact of the current interventions that are aimed at treating PAD by focusing on the subclinical and chronic inflammatory process related to atherosclerosis is less clear.

## 3. Inflammation and PAD 

Atherosclerosis is a well-established chronic subclinical process promoted by ectopic lipid deposition within the arterial wall layers and local turbulent shear stress injury. Both described triggers converge on the onset of the local inflammatory process [21]. Moreover, inflammation has been long recognized as not only a consequence of atherosclerosis but also a paramount inciting event for endothelial damage and consequent lipid plaque formation. 

Indeed, incident PAD has been shown to increase in subjects with underlying systemic inflammation, highlighting the role of the latter as a component cause and not a mere outcome [22,23].

At the base of this lies atherosclerotic plaque formation, initially triggered by LDL accumulation in the intima and subsequently oxidized by reactive oxygen species [24]. This causes both an increased uptake of oxidized LDL by macrophages and intracellular stress mediated by cholesterol crystal deposition, to which the inflammasome pathways respond by increasing the levels of IL-1 [25].

Activated inflammatory pathways are reinforced by multiple comorbidities affecting subjects with PAD, such as diabetes [26], chronic kidney disease (CKD) [27,28], smoking, a sedentary lifestyle, [29] and, emerging in recent years, microbiome dysbiosis [30,31]. 

The inflammatory process associated with atherosclerosis is supported by cell-mediated pathways, including B lymphocytes, T lymphocytes, and macrophage cells, as well as their soluble products [32]. Individuals with higher serum levels of inflammatory markers associated with this pathway, the most commonly tested being highly sensitive C-reactive protein (hsCRP), are more likely to progress into accelerated forms of atherosclerosis, including CLTI, and overall worse clinical outcomes [5]. 

In addition, several serum biomarkers have emerged as potential candidates to identify patients at a higher risk of developing PAD or progressing to more advanced stages of the disease. In particular, a panel of inflammatory biomarkers [33] was observed to predict the risk of major adverse cardiac events (MACEs) and major adverse limb events (MALEs) at 12 months after lower limb endovascular revascularization (LER) [5], along with an increased reintervention rate before the first year after endovascular therapy [34]. 

As seen above, inflammation is an inciting event of the atherosclerotic process in PAD and plays a paramount role not only in initiation of atherosclerosis, but also in arterial plaque progression and vulnerability [35]. Therefore, the treatment of the mechanisms sustaining the inflammatory process is of utmost relevance for the optimal management of PAD (Table 1). 

Although promising evidence is present in the current literature, supported by preclinical and observational data from subgroup analysis, adequately powered trials are needed to confirm the actual clinical benefits of the anti-inflammatory agents in PAD.

Moreover, the outcomes registered up to now did not show any significant improvement, which in part may be explained by the right population having not being identified yet.

The further development of new therapies is another promising endeavor, although most studies focus on CAD. As with most other beneficial interventions, the former will be studied in PAD in the future. 

For this reason, given the potential benefits provided by the modulation of triggers of inflammation, it is necessary to investigate and understand the anti-inflammatory role of the currently available therapies to further reduce the residual risk associated with chronic inflammation while waiting for a designed target therapy.

## 4. Lipid-Lowering Therapy

### 4.1. Statins 

Statins, one of the most employed lipid-lowering therapies, act by competitively inhibiting the enzyme β-Hydroxy β-methylglutaryl-CoA (HMG-CoA) reductase, devoted to the conversion of HMG-CoA into mevalonic acid, the rate-limiting step in cholesterol synthesis.

The relative deficit in cholesterol sensed by the liver triggers an increased expression of LDL receptors on hepatocyte surfaces, leading to an increased internalization and clearance of LDL.

Statins can decrease LDL levels by up to 55% and significantly reduce the rate of MACEs. As evidenced by Navarese et al., subjects with higher levels of LDL benefited the most the most from statin therapy, and more aggressive treatment provided greater protection [82]. 

Regarding PAD, the target level of LDL-c below 55 mg/dL is currently a Class 1 recommendation. As a first-line therapy, statins are mainly used as hypolipidemic agents, with studies showing an increase in amputation-free survival of up to 56% [36] and a slower decline in walking capacity in patients with PAD [37].

Aside from the traditional mechanisms of action, a vast body of literature exists on the potential pleiotropic effects of statins and their role as modulators of inflammation. 

The first clues of the plausible anti-inflammatory effects of “traditionally non-immunomodulatory” drugs were observed in the JUPUTER study (Justification for the Use of Statin in Prevention: An Interventional Trial Evaluating Rosuvastatin) [83], the IMPROVE-IT trial (Improved Reduction of Outcomes: Vytorin Efficacy International Trial) [84], and Pravastatin or Atorvastatin Evaluation and Infection Therapy-Thrombolysis in Myocardial Infarction 22 (PROVE-IT TIMI 22) [85]. In these three studies, patients who were on statin therapy showed a significant reduction in hsCRP levels compared to the placebo. 

Starting from the culprit cells involved in the atherosclerotic process, Feig et al. demonstrated in animal studies that statins were able to induce the emigration of CD68+ macrophages from plaques by inducing the expression of CCR-7 [38].

On the same note, lovastatin and simvastatin were noticed to suppress inflammation through their interaction with leukocyte function antigen-1 (LFA-1), possibly preventing the attraction of inflammatory cells and the activation of T cells at the site of atherosclerotic plaque [39].

Moreover, although mainly explored in central nervous system autoimmune disorders, statins were shown to induce IL-4-dependent Th2 cell differentiation, with the subsequent secretion of anti-inflammatory molecules [40].

Moving from cellular mechanisms to molecular ones, statins have been shown to interfere with isoprenoids, such as Rho and Ras GTPase. More specifically, statins exert an inhibitory effect on these signal transduction pathways, which are implicated in cell trafficking, motility, and proliferation. A change in the actin cytoskeleton brought by this inhibition may not only affect atherosclerotic plaque, but can also decrease smooth muscle contraction [86]. It logically follows that claudication symptoms may be reduced with statins. 

A further mechanism underlined once again by the inhibition of the isoprenoid pathway is the upregulation of NO, which may further promote vasodilation and the subsequent symptomatic relief of claudication [41].

Lastly, statins decrease PAI-1 expression while increasing the levels of tissue plasminogen activator at the same time, which can potentially protect against CLTI altogether [87] (Table 1).

All of these anti-inflammatory properties result in increased plaque stabilization while reducing the risk of the rupture of vulnerable plaques [88,89]. 

As a concluding remark, statins could be beneficial not only as a preventive measure in the initial stages of the disease by preventing inflammatory cells’ attraction and thus atherosclerotic plaque progression, but also in later stages for symptomatic treatments by mediating vasodilation and decreasing the long-standing chronic inflammation underlying PAD.

### 4.2. Ezetimibe

Ezetimibe inhibits the reabsorption of cholesterol from the intestine, mediated by Niemann-Pick C1-Like 1 protein [90], decreasing the amount of cholesterol delivered to the liver. This causes an increased expression of LDL receptors for the liver to compensate for the unfamiliar environment established [42]. 

Prescribed both as a monotherapy or a fixed-dose combination with other lipid-lowering agents, mainly statins, ezetimibe lowers the incidence of sudden cardiac death, myocardial infarction, and coronary artery revascularization, although no difference in all-cause mortality has been observed [91].

In a secondary analysis of the IMProved Reduction of Outcomes: Vytorin Efficacy International Trial (IMPROVE-IT) [43], ezetimibe reduced cardiovascular risk in patients with polyvascular disease, with the greatest benefit obtained by high-risk classes, including PAD of the lower limbs. Of notice, the medication reduced LDL cholesterol by only 14 mg/dL on average, and no more than 19.4% at doses of 10 mg [42], hinting toward different off-target effects. 

Focusing on anti-inflammatory properties, preliminary cellular studies have shown the downregulation of TNFα gene expression, possibly by affecting NF-κB [44]. Moreover, the use of ezetimibe upregulates metabolic pathways, including phosphorylated glycogen synthase kinase and Akt, involved in keeping the mitochondrial membrane potential, thus preventing apoptosis, with a net reduction in endothelial damage-induced stress [45].

Additional in vitro studies investigating inflammatory cytokine panels found that ezetimibe alone was able to reduce only IL-1β expression after 25-Hydroxycholesterol administration, whereas effects on IL-18 and IL-35 were observed only in combination with statins. Similarly, anti-inflammatory TGF-β levels did not significantly change with ezetimibe alone [92].

This effect on inflammatory cytokines may also be partially explained, as evidenced by Cho et al., by a reduction in adipocyte size and secretory activity in visceral adipose tissue [93].

Of note, when analyzing human studies, including the PAD population [94], most of the works published evaluate ezetimibe in combination with statins only; therefore, its effect on the variables investigated is probably limited compared to that of statins [95,96] (Table 1).

Proven cholesterol reduction, mediated by the adjunctive use of ezetimibe together with statins, helps to reduce vascular inflammation. However, further studies are needed to accurately assess the anti-inflammatory role of ezetimibe in atherosclerosis and in PAD of the lower limbs.

### 4.3. PCSK9 Inhibitors 

Proprotein convertase subtilisin/kexin 9 (PCSK-9) mediates the hepatic degradation of LDL receptors by modulating LDL-c clearance and reducing its serum levels. PCSK9-targeting monoclonal antibody inhibitors (PCSK-9i) have been shown to greatly facilitate the achievement of LDL-c goals by significantly improving outcomes related to atherosclerotic burden [97,98]. The lipid-lowering effect seems to be the most evident, despite it not being the only mechanism of action.

Although CRP, IL-6, IL-1b, and TNF-a are some of the most common and reliable markers of systemic inflammation, there are controversial opinions regarding their sensitivity to the inflammatory profile of patients on PCSK9-I [99]. However, the role of PCSK9 on inflammation has been fully established, as this molecule participates in the whole process of atherosclerotic plaque formation and modulates local arterial wall inflammation. 

In fact, circulating PCSK9 produced by the liver contributes to the activation of macrophages via the LDL receptor scavenger pathway and the induction of foam cell formation, cytokine release, and monocyte recruitment, resulting in further atherosclerotic plaque growth [46]. Furthermore, PCSK9 regulates cholesterol efflux into macrophages through the inhibition of the ATP-binding cassette transporter that modulates the macrophage content of oxidized cholesterol [47] and affects the ectopic deposit of foam cells within the intima wall [100].

PCSK9 is also involved in vascular remodeling, regulating endothelial cell apoptosis and the differentiation of VSMCs into the collagen-producing phenotype that contributes to atherosclerotic plaque growth and vulnerability [46].

The TLR4/NFkB pro-inflammatory signaling pathway is involved in the modulation of the immune system and has been observed to be crucial in the development and progression of vascular atherosclerosis. PCSK9 overexpression stimulates immune and endothelial cells to initiate a cytokine cascade by stimulating the TLR4/NFkB pathway [48]. PCSK9 inhibition, on the other hand, attenuates oxidative toxicity, induced by hydroperoxides and malondialdehyde produced by the pro-inflammatory milieu [101,102] (Table 1).

In addition to the brilliant LDL-c reduction results documented in the Evaluation of Cardiovascular Outcomes After an Acute Coronary Syndrome During Treatment With Alirocumab (ODYSSEY OUTCOMES), Further Cardiovascular Outcomes Research With PCSK9 Inhibition in Subjects With Elevated Risk (FOURIER), and, most recently, the Trial to Evaluate the Effect of ALN-PCSSC Treatment on Low Density Lipoprotein Cholesterol (ORION), PCSK9i could be an effective strategy to reduce LDL-c retention within the arterial wall and attenuate the inflammatory process taking place directly on the atherosclerotic plaque. The pleiotropic effects of PCSK9i remain underreported, but may explain the further reduction in MACEs and MALEs, independently of serum LDL-c levels.

## 5. Antithrombotic Therapy

### 5.1. Cilostazol and Prostaglandin 

Cilostazol was, for many years, the only dedicated medication for patients with PAD. This quinolone derivative inhibits phosphodiesterase III, increasing the availability of cAMP to promote vasodilation through the inactivation of myosin light chain kinase, as well as platelet aggregation [103], given the increased levels of platelet activation that is observed in PAD patients [104].

In animal studies, cilostazol has been demonstrated to have anti-TNF activity [49,50]. Moreover, its anti-inflammatory effects have also been explored through decreased levels of the expression of VCAM-1, MMP-9, and PAI-1, hinting not only to the simple symptomatic relief of claudication solely through vasodilation, but also to a slow, underlying process of inflammatory modulation. Significant angiogenesis was also observed in mice treated with cilostazol [105,106], with increased expression of eNOS and CD31+ cells [51]. 

In humans, at a twice-daily oral dose of 100 mg, cilostazol improves walking distance in people with intermittent claudication, although there is no sufficient evidence about its effectiveness in preventing amputation and revascularization [107].

Indeed, an increased intermittent claudication distance of 35% and an increase in maximum distance walked of 41% were observed in the cilostazol-treated group [108].

In a study by Hsieh et al., conducted on diabetic patients with intermittent claudication, cilostazol significantly decreased their levels of hsCRP and adiponectin [52], suggesting a further modulatory role on inflammation and metabolism. 

As noted by Lessiani et al., iloprost, a structural analog of prostacyclin, has known anti-inflammatory, antithrombotic, and vasodynamic effects that can be similarly mimicked by cilostazol. In fact, the inhibition of platelet function and the improvement in endothelial cell functions is mediated by a selective inhibition of the phosphodiesterase type-3 molecular pathway, with a consequent increase in cAMP production. Indeed, a decreased level of soluble CD40L was observed, accompanied by increased plasma nitrates. Moreover, iloprost treatment reduced residual thromboxane synthesis [53] (Table 1).

To summarize, both cilostazol and prostaglandin analogues have shown not only short-term symptomatic relief, but also a more silent but effective process of inflammatory modulation.

### 5.2. Antiplatelets 

Platelets play a critical role in thrombosis, and antithrombotic therapy targeting platelet activation and aggregation is strongly recommended in the current cardiovascular guidelines on polyvascular disease. Aspirin and clopidogrel are the most recommended antiplatelet treatments, and strong scientific knowledge support their use in symptomatic PAD for the prevention of MACEs [109]. Although a positive role in MACEs has been observed, the use of antiplatelet therapy alone may protect only slightly against lower limb events and the progression of PAD [110,111].

Aspirin causes the irreversible inactivation of cyclooxygenase (COX) enzyme types 1 and 2 by acetylating its serine residue. Thus, aspirin causes a reduction in the synthesis of prostaglandins and thromboxane. The main effect is a significant reduction in pro-inflammatory and prothrombotic effects, mediated by some isotypes of prostanoids. Furthermore, COX2 acetylated by aspirin produces lipoxins that foster an anti-inflammatory effect. The antithrombotic effects of low doses of acetylsalicylic acid are mediated by the ability to largely reduce the formation of thromboxane A2 in platelets, while the synthesis of anti-inflammatory prostanoids (e.g., prostaglandin I2) is only minimally affected. Additionally, aspirin-modified COX-2 contributes to further increase the expression of lipoxins and prostacyclins, with a net anti-inflammatory and anti-thrombotic effect [112]. 

Therefore, the downregulation of prothrombotic factors relative to prostacyclin expression results in the well-known, beneficial effects of long-term, low-dose aspirin therapy [113].

In any pro-inflammatory condition, such as a peroxidized plaque, prostanoids are produced and induce local vasodynamic changes; the recruitment and infiltration of leukocytes in the layers of atherosclerotic plaques, where macrophages can be activated; and increased cytokine production by overexpressing NF-kB [114].

The inflammation that promotes atherogenesis appears to be mediated by COX-1 products such as thromboxanes, prostacyclin (PGI2), and prostaglandin E_2_ (PGE2), which appear to be the main mediators of the development of atherogenesis. Indeed, microsomal prostaglandin E synthase-1 (mPGES-1) is a perinuclear protein that is overexpressed in atherosclerotic plaques and contributes to PGE2 production, as well as accelerating the growth of macrophage-rich plaques. Aspirin significantly reduces this vicious process through the early inhibition of this pro-inflammatory cascade, preventing the formation of the atherosclerotic inflammasome [115]. Furthermore, since aspirin can exert a multisite blockade of the COX pathways (in addition to those of platelets), it hinders any redirection to other sites of thromboxane production, such as intraplaque macrophages and VSMC [116].

Clopidogrel is a second-generation thienopyridine and a widely prescribed prodrug with anti-thrombotic activity through the irreversible inhibition of the P2Y receptor on platelets. Clopidogrel remains a mainstay of PAD treatment, with a significant reduction in MACEs in this population [117].

The beneficial effect of clopidogrel in PAD could be explained by the stabilization of arterial plaque, mediated by a reduction in the local cytokine burden.

Bacterial lipopolysaccharide-mediated inflammation induces a cytokine storm, and clopidogrel has been shown to decrease levels of CRP, IL-6, and TNF-a, known promoters of plaque vulnerability [55]. Therefore, in advanced stages of PAD, characterized by ulcers and gangrene, clopidogrel may exert an additional protective role against MACEs and MALEs.

Moreover, recent studies have suggested that P2Y12 receptor activation promotes VSMC-derived foam cell proliferation, which is an important step in the development of atherosclerosis [118]. 

In PAD patients, short administrations of clopidogrel can reduce the expression of adhesion molecules such as p-selectin and CD63, as well as regulate upon activation, normal t cell expressed and presumably secreted (RANTES–CCL5), modulating the adhesion of platelets [56] and the recruitment of leukocytes into vessel walls [119] (Table 1).

Antiplatelet therapy remains an essential therapy, exerting functions beyond the regulation of platelet adhesion, and solid anti-inflammatory benefits provide additional protection in PAD.

### 5.3. Low-Dose Rivaroxaban

The Cardiovascular Outcomes for People Using Anticoagulation Strategies (COMPASS) and Vascular Outcomes Study of ASA Along With Rivaroxaban in Endovascular or Surgical Limb Revascularization for PAD (VOYAGER PAD) studies demonstrated the efficacy of dual pathway inhibition (DPI), a recent antithrombotic strategy that is intended to modulate the pro-thrombotic activity of platelets along with the direct inhibition of pro-coagulant factors in patients with PAD. In fact, the combination of anticoagulant and antiplatelet therapies has proven to be an extraordinarily effective strategy to significantly reduce the incidence of cardiovascular and limb events, including ischemia and limb amputation. Adding low-dose rivaroxaban to aspirin therapy has opened new opportunities to counteract the progression and incidence of the vascular complications of PAD, with a non-relevant risk of fatal or critical organ bleeding [120].

Rivaroxaban is a direct selective inhibitor of activated coagulation factor X (FXa), which is a serine protease involved not only in hemostasis, but also in various inflammatory processes. These include the activation of protease-activated receptors (PARs), particularly PAR1 and PAR2. PARs belong to a subfamily of related G protein-coupled receptors, which are predominantly expressed in platelets, leukocytes, endothelial cells, and VSMCs. During the coagulation cascade, these receptors are sensitive to FXa, which cleaves their extracellular domain, promoting endothelial cell activation, hemostasis, and vascular tone [121].

The PARs expressed on endothelial cells contribute positively to the initiation of the cellular-mediated mechanism underlying the inflammatory process by upregulating endothelial adhesion molecules such as VCAM-1, ICAM-1, and E-selectin. These proteins are implicated in the permeability of the endothelial barrier, facilitating the interaction between endothelial cells and circulating leukocytes [58,59].

Furthermore, FXa can affect the cytokine pathway by stimulating PAR activation on endothelial cells, which induces the overexpression of pro-inflammatory cytokines such as IL-1, IL-6 (by VSMCs), and IL-8 (by leukocytes), promoting atherosclerosis and thrombosis [122].

Rivaroxaban can exert direct anti-inflammatory effects on vessel walls. The inhibition of FXa activity reduces PAR-2 cleavage, resulting in less superoxide anions being produced by NADPH oxidases and increased eNOS expression. The latter leads to the rapid scavenging of NO, which plays a leading role in vascular health, antiplatelet function, thrombosis, vascular inflammation, and atherosclerosis [60].

Evidence demonstrating the effects of rivaroxaban on athero-inflammation has shown further pathways in which FXa is involved. In particular, the reduced translocation of NF-kB to the nucleus that is mediated by FXa inhibition leads to a significantly lower expression of pro-inflammatory genes. Moreover, rivaroxaban modulates the leukocyte–platelet–endothelial cell interaction by preventing the formation of microthrombi and the activation of macrophages in damaged vessels [122].

Thus, in line with recent studies showing the pleiotropic effects of rivaroxaban on endothelial calcification and inflammation, the modulation of the coagulation cascade, particularly by targeting factor Xa [123] with therapeutic doses of rivaroxaban, may reduce the inflammation that promotes the progression of atherosclerotic plaque (Table 1).

Rivaroxaban is an effective strategy to prevent MACEs and MALEs in PAD and to counteract the mechanisms underlying the initiation and promotion of thrombo-inflammation, even in distal lesions that are often unsuitable for invasive revascularization [124].

## 6. Antidiabetic Therapy

### SGLT2-I and GLP1-RA

Sodium-glucose transporter 2 inhibitors (SGLT2-i) target proximal tubular sodium and glucose reabsorption. The inhibitory effect on the transporter lowers the threshold for glycosuria, allowing a greater amount of glucose to be excreted. 

In recent years, gliflozins have gained increasing importance in the management of chronic heart failure (CHF), diabetes mellitus (DM), and CKD [125,126,127,128], and their role in atherosclerotic diseases is also being explored. 

In fact, in animal studies on ApoE-/- mice, evaluating several parameters of plaque vulnerability, including monocyte chemoattractant protein-1 (MCP-1), macrophage population, and necrotic core size, Chen et al. showed a relevant stabilization of atherosclerotic plaque in mice treated with SGLT2-i, hinting toward a protective mechanism directly acting at the lesion site [129]. In line with the promising results on the effects of gliflozins on atherosclerotic disease, the use of SGLT2-i is associated with a robust reduction in myocardial infarction, stroke, and cardiac mortality, regardless of the ongoing therapy [130]. Furthermore, along with anti-atherosclerotic effects, gliflozins have shown a significant improvement in coronary artery flow [131], showing further benefits on vessels. 

In studies investigating the effects of SGLT2-i on diabetic patients, including the PAD population, worrying evidence of a higher rate of amputations was documented in those treated with canagliflozin, as shown in the CANagliflozin Cardiovascular Assessment Study (CANVAS) trial [132]. However, subsequent new findings on gliflozins, especially empagliflozin, have allayed this first concern by demonstrating that SGLT2i significantly reduces the incidence of MALEs, as pointed out by Subodh et al. [133,134].

In addition to the solid role of SGLT2-i as a first-line therapy for DM, CKD, and CHF, gliflozins also contribute to slowing the progression of atherosclerosis and its vascular complications by modulating the inflammatory process.

The first target is a reduction in oxidative stress by decreasing the activation of the renin–angiotensin–aldosterone system (RAAS). It is known that Angiotensin II initiates the inflammatory cascade by enhancing the expression of nuclear factor-kappa B (NFkB) and promoting oxidative stress and endothelial dysfunction [61], but it also increases the expression of SGLT1 and SGLT2 molecules [135].

Inhibiting the latter transporter, sodium delivery at the macula densa of the distal convoluted tubule is increased, leading to lower angiotensin II levels, reduced ROS production [62], and the upregulation of endothelial nitric oxide synthase (eNOS), as well as nitric oxide (NO) formation, on vessel walls [63]. 

A second effect focuses on the suppression of weight gain and a reduction in pro-inflammatory adipocytokine production. Although shown in mice, empagliflozin triggers the browning of adipose tissue and decreases the mean adipocyte size, along with the modulation of mRNA coding for fatty acid synthesis enzymes [64].

Another paramount target of SGLT2-i is the inflammasome pathway, which is activated upon the intracellular accumulation of cholesterol, the culprit of atherosclerotic disease. Preliminary animal studies have demonstrated that SGLT2-i triggers the upregulation of AMP-activated protein kinase (AMPK), which inhibits nucleotide-binding domain, leucine-rich-containing family, pyrin domain-containing-3 (NLRP3)/apoptosis-associated speck-like protein containing a CARD (ASC), and NF-κB activation, resulting in a net reduction in IL-1β and IL-6 production [136,137,138].

These anti-inflammatory effects were also proven in human studies, where, as shown by Grotta et al., lower levels of circulating IL-6 and uric acid (a known mediator of pro-inflammatory cytokine expression) were found [65].

A further mechanism to be proposed may be the polarization of macrophages toward an M2 phenotype, typically involved in the modulation of inflammatory triggers and the suppression of the inflammasome, although further research is needed on the topic [66,139].

A final target of SGLT2-I is the anti-inflammatory sirtuin pathway, involved in transcriptional regulation, chromatin function, DNA integrity, and repair [140], which is upregulated by gliflozin therapy [141].

Another pharmacological class that was recently introduced as a mainstay of diabetes treatment, especially in patients at high cardiovascular risk, is glucagon-like peptide 1 (GLP-1). The mechanism of action of GLP-1 is mainly exerted by stimulating insulin secretion after oral glucose intake, thanks to the incretin effect. Additional effects include improved cardiac function along with lowered cholesterol and blood pressure [142].

Likewise, for SGLT2i, GLP-1 is involved in the sirtuin pathway. Indeed, as proposed by Balestrieri et al., the administration of the former is associated with increased SIRT6 expression and consequent collagen levels, promoting plaque stabilization [67].

The added effects of GLP1 in targeting atherosclerosis more specifically are summarized in the meta-analysis by Song et.al, which suggests the preventive anti-atherosclerotic effects of GLP1-a, supported by a statistically significant association with a reduction in plasminogen activator inhibitor, LDL, triglycerides, and hsCRP. 

Moreover, a significant increase in normalized flow-mediated dilation was seen in the initial phase of therapy, although significance was lost with time. This may imply that GLP-1 could prove to be a promising strategy to prevent early atherosclerotic vascular complications in PAD of the lower limbs. While exploring the activity of GLP-1 on cytokines, no statistically significant associations with levels of IL-6 and TNF-α were observed [33].

Lastly, GLP1 stimulates endothelial cell proliferation, activating the metabolic pathway PI3K-Akt [143], and, interestingly, it decreases matrix metalloproteinase (MMP) mRNA levels and transcription. MMP-1, MMP-2, and MMP-9 were significantly lower, further contributing to the protection against plaque vulnerability [144].

In conclusion, both SLGT2-i and GLP-1 target inflammatory pathways acting in principle at the endothelial level, promoting NO production and subsequent vasodilation, with GLP1 being especially beneficial in the first months of therapy (Table 1). Whilst the former compound decreases the traditional inflammatory interleukins, the latter acts mainly on plaque stabilization by decreasing the levels of matrix metalloproteinases.

Overall, both play a paramount role in regulating inflammatory processes and, being extensively included in new guidelines, further studies will be needed to better characterize and assess the molecular mechanisms behind these beneficial effects.

## 7. ACEi/ARBs/ARNI 

Angiotensin-converting enzyme inhibitors (ACE-i), angiotensin receptor blockers (ARBs), and angiotensin receptor neprilysin inhibitors (ARNI) are all medications widely employed in the cardiovascular field for a variety of conditions (hypertension, proteinuria, and heart failure), and all target the renin–angiotensin–aldosterone system (RAAS) and neprilysin.

Their mechanism of action slightly differs based on the step affected in the RAA cascade. 

Starting from the most upstream process, ACE-i reduces the conversion of the decapeptide angiotensin I, with limited biological activity, into the octapeptide angiotensin II (ATII), a potent vasoconstrictor, pro-inflammatory mediator, and cardiac remodeler. 

A different strategy can be adopted using ARBs, which, depending on the compound, can either competitively or non-competitively inhibit the ATII receptors and, more specifically, the ATIIR1, mainly involved in myocardial remodeling post-ischemic damage. For this reason, ARBs may be preferred over ACE-i in heart failure patients with a past medical history of myocardial infarction, as the former do not target the other isoform of the ATIIR, ATIIR2, which is instead selectively expressed under inflammatory conditions and mediates the inhibition of cell growth, protection against ischemia, and decreased extracellular matrix formation [145].

Lastly, first approved in 2015 and later indicated for heart failure with reduced ejection fraction (HFrEF), ARNI combines ARBs with neprilysin inhibitors, which on one side benefit from the degradation natriuretic peptides, and on the other maintain low levels of ATII, decreasing its effects on volemia and vascular endothelial function, which would otherwise be elevated in the presence of a neprilysin inhibitor only. 

All the medications listed above indirectly contribute to reducing the atherosclerotic process. Most of the effects are directly promoted by the modulation of ATII activity. 

Polyvascular patients often suffer from hypertension, which is both the cause and effect of atherosclerosis and its major vascular complications. Blood pressure management is strongly recommended in these patients to prevent the incidence and progression of arterial disease, and this drug class is considered a first-line therapy [146].

As previously stated, ATII acts on ATIIR1 that mediates the reduction in cellular reducing power, increased ROS production, and the overexpression of NFkB. Thus, these drugs counteract the ATII-related molecular cascade, which promotes smooth muscle proliferation, leukocyte homing in the intimal layer [68], vasoconstriction due to decreased NO production, and the overproduction of chemokines and adhesion molecules [61,147]. Therefore, the inhibition of ATII activity may be protective against arterial vasoconstriction and inflammation, which are underlying conditions of atherosclerotic vascular disease, including PAD.

The RAAS components are involved in recruiting inflammatory molecules and cells at the site of injury, and once a pro-inflammatory milieu is established, the latter can further generate ATII, propagating the damage. This effect is partially mediated by the increased expression of monocyte chemoattractant protein 1 (MCP-1) and by decreased levels of NO [69].

To this extent, valsartan and olmesartan were studied, and were found to reduce the degree of atherosclerotic burden in polyvascular disease [148,149]. These effects can be extended to the entire class of medication.

Moreover, starting from studies on captopril, ACE-I have been positively associated with a decreased level of oxidative stress, possibly through the scavenging of peroxynitrite [150].

As most of the antihypertensive effects are achieved through a decrease in peripheral vascular resistance (PVR), the integration of these medications in the treatment of PAD patients is of particular importance [151]. Indeed, patients on ACE-i had significant improvements in their maximum walking distance and pain-free walking distance with a delay in PAD progression, although no significant change in the ABI was noted [152] (Table 1). 

The cardiovascular protection exerted by ACEi, ARBs, and ARNIs may also be mediated by their anti-inflammatory effects, observed independently of blood pressure reduction [153].

## 8. Revascularization Procedure

In the late stages of the disease, restoring blood flow to ischemic tissues with invasive procedures becomes an urgent and effective treatment. Among the possible revascularization options, endovascular procedures with angioplasty (with or without stenting) have become the first-choice therapy for most patients with PAD. However, balloon inflation and stent placement induce severe arterial wall damage and may promote local inflammation and restenosis. 

Restenosis after vascular procedures is the result of hypertrophic wound healing due to the excessive activation of myofibroblasts and the excessive production of extracellular matrix proteins, leading to the formation of hypertrophic neointima and a narrowing of the vessel lumen. Shear stress and vascular injury caused by angioplasty and stent placement lead to an exaggerated expression of chemokines and interleukins (which promote monocyte and vascular smooth muscle cell (VSMC) recruitment) and adhesion molecules such as E selectins, P selectins, and cell adhesion molecules (CAMs) (facilitating the migration of inflammatory cells) [154].

MMPs contribute to constrictive remodeling, intimal thickening, and revascularization failure [155,156].

Comparing the periprocedural profile of circulating inflammatory markers to admission values in symptomatic PAD patients undergoing elective lower limb percutaneous revascularization, a significant elevation in serum MMP-3 and -7 levels was observed 24 h after intervention.

At present, neither a clear mechanism underlined by the rise in the markers nor the effect of these on vascular inflammation and clinical outcomes (arterial patency) has been observed. Moreover, there is further uncertainty in their use as post-procedure outcome predictors and therapeutic targets [157]. 

Healing progression is tightly related to the ability to synthetize the extracellular matrix, whereas the degradation of the latter, mediated by MMPs, causes delayed or failed wound healing. MMP-2 and MMP-9 might play a key role in this process due to their affinity for basement membrane collagen type IV and laminin [158]. High postoperative levels of MMP-2 and MMP-9 were indeed predictive of wound healing failure in PAD patients, suggesting that the pre-operative modulation of these biomarkers could prevent revascularization complications [159,160].

A notable change in the serum levels of local and systemic mediators (such as CRP, TNF-α, serum amyloid A, bFGF, TGF-β1, IL-10, and fibrinogen) was observed before and after angioplasty [161]. In addition, the intensity of the inflammatory process is also reflective of the extent and complexity of the revascularization due to the necessity of a more aggressive manipulation, resulting in a higher occurrence of vessel remodeling and lumen occlusion [162].

Finally, together with local mediators, genetic factors also contribute to post-intervention inflammation and revascularization failure [163,164,165]. The expression of heme oxygenase-1 modulates several inflammatory pathways (including IL-6) with direct inhibition of VSMC activity [154]. In addition, several genes (the extracellular signal-regulated kinase (ERK) gene pathway has the greatest relevance) have been found to regulate leukocyte extravasation, angiogenesis, cell proliferation, and Toll-like receptor (TLR) 2 activation in response to oxidative stress and local cytokine storms promoted by iatrogenic vessel damage [161,166,167] (Table 1).

Although no clear consensus has been reached on which among the biomarkers studied are the most promising for PAD, the monitoring of periprocedural inflammatory biomarkers may prove to be a novel predictor of clinical outcomes for patients with PAD.

## 9. Diet and Dietary Habits 

Diet remains the most cost-effective and successful way to prevent and treat CVDs (cerebrovascular and cardiovascular diseases), including PAD. However, despite nutrition potentially being a strategy for the atherosclerotic pandemic, it remains the most under-prescribed and underrated therapeutic approach. Appropriate nutritional programs, that include diets tailored to each individual need, should also be accompanied by a thorough investigation of dietary behaviors that further contribute to the unhealthy relationship between PAD and nutrition [168].

Although obesity is a predisposing condition for the development of PAD, malnutrition and sarcopenia in PAD are associated with more aggressive disease progression and a higher amputation rate. This obesity paradox can be explained by the stronger pro-inflammatory phenotype that occurs with the development of sarcopenia that accelerates the progression of PAD [169].

A diet with a high intake of proteins of animal origin impairs the lipid profile with an increase in circulating oxidized LDL-c, which favors the development of reactive atherosclerotic plaque that is rich in foam cells. On the other side, plant stanols, flavonoids, and phytosterols contained in plant-based diets partially recover the endothelial dysfunction and mitigate the macrophage overproduction of TNF-α, IL-6, IL-1β, and chemokines [70,71,72].

Micronutrients play a critical role in the modulation of low-grade inflammation in PAD. Vitamins A, C, and E contribute to delaying the risk of rupture of vulnerable plaques by modulating intraplaque oxidative stress [35,73,74], while B vitamins reduce the proliferation of VSMCs, which affect arterial remodeling and promote pro-thrombotic inflammation [74]. The healing effects on ischemic ulcers and immune modulation during ischemic triggers of vitamin C and D supplementation, respectively, contribute to a reduction in MACEs and MALEs [170,171]. Other micronutrients (such as zinc and magnesium) are crucial for maintaining the homeostasis of immune cells [172]. Polyunsaturated fatty acids (PUFAs) are mainly derived from fish and seeds. In addition to bringing benefits on the metabolism, PUFAs can influence the outcomes of the revascularization of limbs through a modulation of the inflammatory flare following angioplasty-mediated vessel injury [173]. PUFAs stabilize macrophage cell membranes, reduce LDL-c oxidative stress, decrease NF-kB expression [75], and reduce leukocyte adhesion via the expression of vascular cell adhesion protein 1 (VCAM-1) [76].

Soluble fiber intake promotes intestinal eubiosis, documented by an increased production of short-chain fatty acids that reduce systemic inflammation [77] by modulating PPAR-α activity [174], which controls the β-oxidation of peroxisomal/mitochondrial fatty acids during ischemic injury and oxidative stress [78].

So far, there is no agreement on the most appropriate diet for PAD. Thus, several options have been investigated. The Mediterranean diet is the most recommended dietary pattern in CVD with a balanced proportion of healthy nutrients, resulting in protection against the development of those risk factors underlying metabolic syndrome, which contribute to maintain low-grade systemic inflammation [175].

The large intake of phenolics and PUFAs from vegetarian and vegan diets appears to have the ability to reverse atherosclerotic plaque formation in PAD by mitigating trimethylamine-N-oxide (TMAO)-mediated vascular inflammation [79] and promoting the clearance of ectopic oxidized lipid deposits and foam cells [176].

Ketogenic diets [177] and intermittent fasting [178] affect metabolism and gut microbiota, leading to significant weight loss through the degradation of adipose tissue. The change in body composition is associated with a net decrease in cytokine production from fat, responsible for the hyperactivation of macrophages and lymphocytes in atherosclerotic plaque, triggering plaque vulnerability [177,179].

Finally, another aspect of nutrition that remains poorly understood is the behavioral, emotional, and psychological dimension associated with food [180,181]. An irregular frequency of meals, emotional eating, a high rate of skipping meals [182], night eating [183], and an unbalanced fast food/home cooking ratio [184] are an interesting new field to further explore in order to understand the effects of “how we eat”, as well as “what”, on individual metabolism and inflammatory profiles [180] (Table 1).

## 10. Supervised Exercise Therapy (SET)

Muscle wasting and myosteatosis are the early structural abnormalities that patients with PAD develop as the disease severity increases. The resulting disability is responsible for worse outcomes, functional limitation, and impaired systemic homeostasis, as muscles play a critical role in metabolism and inflammation [185]. The importance of noninvasive management over revascularization is supported by a marked reduction in the incidence of vascular adverse events, costs, and hospitalization [186,187]. Individualized supervised exercise in the early stages of PAD significantly increases ambulation performance by delaying the otherwise unavoidable need for revascularization [188]. Additionally, adding a home exercise program to standard medical care [189] results in increased vessel patency after revascularization [190]. SET in metabolic syndrome is the most natural and effective strategy to lose excess fat and restore impaired metabolism. Improvements in the lipid profile, glucose metabolism, and insulin resistance are observed regardless of the intensity of weight loss [80]. Indeed, a profound functional and metabolic change in visceral and abdominal fat is observed early during SET, with a marked reduction in oxidative stress [191]. After only 8 weeks of SET, patients with PAD experience a remarkable relief from claudication with an increase in walking performance [192], which is directly correlated with a reduction in endothelium-derived inflammatory markers (E-selectin and ICAM-I) [81] (Table 1).

## 11. Immunomodulatory Therapy: Promising Perspectives 

Although immunomodulatory drugs are not yet included in guideline-directed medical therapy, a growing body of evidence predicts their future use in the therapeutic management of patients with PAD. 

More specifically, the Canakinumab Anti-Inflammatory Thrombosis Outcome Study (CANTOS) and The Cardiovascular Inflammation Trial (CIRT) enrolled patients with established CVD and studied the effects of canakinumab and methotrexate, respectively, in addition to the best medical therapy. 

Interestingly, both medications significantly reduced the median levels of inflammatory biomarkers, and although methotrexate was not equally protective, the canakinumab-mediated inhibition of IL-1β, IL-6, and hsCRP provided additional protection against MACEs [193].

In view of these initial positive results on the possible adoption of immunosuppressive therapy in atherosclerotic disease, other studies have emerged in recent years. 

Tocilizumab is considered a mainstay in the treatment of rheumatological disorders sustained by the activation of the IL-6/IL-6R pathway, and although its use is associated with an increase in cholesterol levels, it appears to have a safe CV profile [194] and to be protective against vascular diseases, as it improves endothelial function with a reduction in arterial stiffness [195]. 

IL-1 signaling mediates a cascade of pro-inflammatory cytokines with the activation of caspase-1 and the NLRP3 inflammasome. Anakinra targets the IL-1 pathway by interrupting the damage of the sterile autoinflammatory process that is highly expressed in atherosclerotic plaques. Therefore, in view of its safety, it is being considered for other non-traditional inflammatory diseases including atherosclerosis and vascular diseases [196]. 

Furthermore, anakinra-mediated IL-1 inhibition is associated with a marked improvement in vascular function due to a reduction in oxidative stress and thrombosis mediated by nitro-oxidative reagents and endothelin [197]. 

Inclacumab binds directly to P-selectin on the surface of endothelial cells. The inhibition of the latter blocks the pro-inflammatory and prothrombotic effects mediated by this cell adhesion molecule [198]. The overexpression of P-selectin in PAD is an independent risk factor and is associated with the disabling complications that follow disease progression [199]. 

Meanwhile, a Phase 3 research study into the effects of ziltivekimab (ZEUS) to evaluate cardiovascular outcomes in participants with established atherosclerotic CVD (ClinicalTrials.gov Identifier: NCT05021835) is ongoing. Ziltivekimab is an IL-6 inhibitor that has emerged as a promising therapy for reducing the incidence of atherosclerosis-related thrombosis [200]. 

Finally, the Colchicine Cardiovascular Outcomes Trial (COLCOT) also demonstrated a favorable role of immunomodulatory therapy in atherosclerotic patients. The use of low-dose colchicine, an old and inexpensive anti-inflammatory drug, can be an adjunctive therapy in selected patients characterized by a more prominent inflammatory profile [201]. 

Based on these findings, while not yet strongly recommended for the protection of cardiovascular diseases other than chronic inflammatory pathologies, new ongoing studies are investigating whether purely anti-inflammatory drugs could provide additional benefits in other high-risk populations such as PAD patients. The inflammation expressed in PAD appears to be driven by the inflammasome-IL axis, which would make these patients more sensitive to drugs that modulate IL-1β activity (canakinumab), as well as the NLRP3 inflammasome, which responds to oxidative stress by initiating the cytokine cascade of IL-1β and IL-18 (colchicine).

Therefore, although neither canakinumab nor colchicine have been shown to reduce mortality, in view of the latter’s safer infectious profile, low-dose colchicine has been considered a promising treatment for PAD [6]. Indeed, Low Dose Colchicine in Patients with Peripheral Artery Disease to Address Residual Vascular Risk (LEADER-PAD) is a randomized, double-blind, multi-center pilot study with an estimated completion date of September 2024 that is investigating the safety and vascular outcomes of patients with PAD taking low doses of colchicine (ClinicalTrials.gov identifier: NCT04774159 accessed on 27 October 2023).

## 12. Discussion

### 12.1. The Residual Cardiovascular Risk

PAD is defined as an important risk factor for cardiovascular (CV) adverse events and prognosis. Therefore, patients with PAD are allocated the highest risk class for atherosclerotic complications. The pandemic increase in the incidence and prevalence of CVD has progressively led to more stringent targets of traditional risk factors *(2021 ESC (European Society of Cardiology) Guidelines on the prevention of cardiovascular disease in clinical practice)*. Although treatment options and their efficacy in managing underlying CVD predisposing factors have increased, a relevant proportion of people at very high CV risk still does not meet the recommended targets. Patients with PAD represent the most undertreated population [202]. Since the introduction of CV prevention guidelines, the recommended goals have never been easily achieved and a relevant percentage of patients have recurrent CV events before starting the most modern therapies. This suboptimal strategy has a significant impact on healthcare costs and has consequences in terms of disability and mortality [203].

However, the early introduction of advanced therapies represents an emerging and promising alternative to the current goal-based prescription of drugs. There is a growing list of pathologies where the introduction of the latest therapies is permitted from the earliest stages of the disease, regardless of the achievement of the surrogate biomarkers that are used thus far as drivers of treatment management. For example, GLP1 agonists and SGLT2 inhibitors are now considered as first-line therapies along with metformin for the treatment of diabetes mellitus, and should be started soon after diagnosis [204]. On the same note, the early introduction of ARNI and SGLT2 inhibitors may be considered in patients with reduced ejection fraction [205], and PCSK9i can be prescribed immediately after an acute myocardial infarction [206].

These new therapeutic approaches have been supported by the overwhelmingly positive results observed. The latter are probably explained by the still poorly explored pleiotropic effects providing additional benefits.

So far, the prevention of atherosclerosis and the treatment of vascular complications have focused on traditional risk factors, and a surrogate biomarker along with recommended targets have been defined for each. Each drug was thought to have one mechanism of action or pathway and be responsible for only one effect.

However, despite the optimal management of all these factors (single pathways), a substantial proportion of patients with PAD still has recurrent events and worse outcomes, suggesting underlying conditions that, if left untreated, may accelerate the progression of atherosclerosis.

The particularly aggressive phenotype in these patients may be explained by the existence of the “Residual Risk”, defined as that risk which persists despite optimal therapy [207]. 

Currently, three pathways of residual cardiovascular risk and corresponding specific treatments have been identified. The first group is the “residual thrombotic risk”, which can be managed with low-dose rivaroxaban (COMPASS) and dual antiplatelet therapy (PEGASUS, TIMI54, and THEMIS-PCI). The second group is the “residual metabolic risk”, which may be managed with Icosapent ethyl (REDUCE-IT), PCSK9i (ODISSEY, FOURIER, and ORION) and new glucose-lowering drugs such as SGLT2i (EMPA-REG, DECLARE TIMI 58) and GLP1a (LEADER). Lastly, and most relevant for this discussion, the third and most complex pathway is inflammation, in particular the residual low-grade systemic inflammatory burden, or “residual inflammatory risk”, that contributes to the progression and vulnerability of atherosclerotic plaques with multiple arterial bed involvement [207,208]. Initially, the higher prevalence of CVD and increased susceptibility to the development and progression of atherosclerosis in patients affected by chronic inflammatory, infectious, and autoimmune diseases suggested the possibility to adopt immunosuppressive therapy for atherosclerosis (ACC Guidelines/ AHA (American Heart Association) for Primary Prevention—2019). Thus, patients treated with immunomodulatory therapies favored new studies evaluating their outcomes after improvements in their inflammatory profiles [6]. The first evidence came from pioneering experiments, COLCOT, CANTOS, and JUPITER, which demonstrated how colchicine, canakinumab, and statins, respectively, can improve the clinical outcomes of polyvasculopathic patients with a strong underlying inflammatory load. In particular, low-dose colchicine can be considered in patients with a suboptimal control of risk factors or in patients with recurrent CV events despite optimal therapy. This has opened up a promising hope of attempting the use of colchicine in patients suffering from PAD, in particular to slow down the progression of the disease, reduce the incidence of MACEs and MALEs, and prevent the failure of lower limb revascularization (ClinicalTrials.gov identifier: NCT04774159).

### 12.2. Current Available GDMT to Manage the Residual Inflammatory Risk

To date, there are no adequately powered studies evaluating the efficacy of direct anti-inflammatory agents in PAD. Furthermore, the anti-inflammatory effects of the current non-immunomodulatory therapies available for PAD have not been studied, but exploring their pleiotropic properties could help in finding new strategies to manage this disease. 

As shown in Table 1, the pleiotropic effects of the currently available GDMT could explain the additional benefits observed in patients’ outcomes, regardless of the achievement of the recommended target levels of the surrogate biomarkers that are used for dose–response assessments. 

Lipid-lowering therapies were the first non-anti-inflammatory drugs to have presented extremely interesting immunomodulatory effects by reducing the serum levels of inflammatory markers with a consequent slowdown of the atherosclerotic process. In particular, statins and PCKS9i have a recognized role in atherosclerotic plaque stabilization by also directly modulating the intraplaque inflammasomes and reducing the oxidative stress within the arterial wall [209]. In addition to lipid-lowering therapy, some antithrombotic drugs, such as aspirin, have also shown a protective effect in terms of lower incidences of MACEs by targeting pro-inflammatory pathways. The modulation of inflammatory factors responsible for the coagulation cascade and platelet activation remains an important support strategy to reduce the thrombotic risk of patients with polyvascular disease [210].

GLP1a and SGLT2i have emerged as innovative and effective therapies to improve dysmetabolism, which is often present in diabetic PAD patients. The extraordinary properties of these drugs can effectively reduce patients’ cardiovascular risk and, interestingly, provide a clear improvement in their inflammatory burden by modulating lipid and glucose metabolism, reducing adipose tissue, and controlling systemic cytokine expression [211]. 

Among classic therapies, ACEi, ARBs, and ARNI play a fundamental role as anti-remodeling drugs with proven efficacy on the cardiovascular system. Moreover, these drugs also show additional benefits on endothelial function and arterial stiffness, counteracting the mechanisms that promote the development of atherosclerosis and vascular inflammation [153]. 

Finally, along with medical therapy, non-pharmacological therapeutic strategies remain the mainstay of PAD management. Every patient should follow a diet and exercise even before taking medications. A growing body of evidence has shown that patients who strictly follow an appropriate dietary program and supervised exercise therapy show a clear slowdown in the progression of atherosclerosis, a reduction in the incidence of macrovascular complications, and a better inflammatory profile, resulting in a less aggressive disease phenotype [168,212].

In the coming years, immunomodulatory treatment will finally become an additional therapeutic strategy for patients with a strong inflammatory burden, markedly represented residual risk factors, or who present disease progression despite optimal medical therapy (European Society of Cardiology, 2022). New ongoing studies will increase knowledge on the benefits of this class of drugs and hopefully provide new therapeutic options for the management of PAD.

### 12.3. Future Perspectives to Manage the Residual Inflammatory Risk

Atherosclerosis remains a systemic process characterized by a low-grade subclinical inflammation responsible for the formation, progression and rupture of atherosclerotic plaques. The humoral and cell-mediated mechanisms that modulate the inflammatory burden are still matter of investigation, but the knowledge of these pathways will allow the introduction of new therapeutic options in the pharmacological management of patients with PAD.

In recent years, other pathways involved in the pathogenesis and progression of PAD have been discovered. Therefore, a panel of novel molecules has been identified that could be used as targets of future therapies and as markers of treatment and prognosis.

Of notice among the most recent serum biomarkers, osteoprotegerin (OPG), a member of the TNF receptor family, has gained attention due to its inhibiting role on the RANK-RANKL signaling pathway, through which arterial media calcification may occur, as preliminarily evidenced in animal studies [213,214].

Although mainly investigated in subjects affected by type 2 diabetes mellitus, OPG is significantly elevated in PAD patients and could have a predictive role in the severity and in the progression to CLI [215,216].

Moreover, a study by Kadoglu et al. suggested a potential predictive role in MACE occurrence after endovascular revascularization [217]. 

A second candidate biomarker is sortilin-1, a membrane glycoprotein involved in lipid metabolism [218], which is increased in PAD patients sampled from the diabetic population and correlates with disease severity [219]. Sortilin-1 is also associated with increased MACEs and MALEs after revascularization, suggesting a possible use as a disease-progression marker.

On the same note, omentin-1, a widely expressed adipocytokine inducing a Th2 anti-inflammatory phenotype [220], is inversely associated with PAD [221], while positively associated with the ABI [222]. Lower levels of omentin-1 are also associated with more frequent MALEs and MACEs 12 months after revascularization [223].

Another marker worth mentioning is fibroblast growth factor 23 (FGF23). This protein is involved in vascular calcification, and significantly higher levels were observed in patients with PAD versus patients without PAD [224], provided that the subjects had a glomerular filtration rate of >60 mL/min [225].

Lastly, Klotho, a ubiquitous protein with both transmembrane and soluble isoforms, can negatively modulate telomerase activity with anti-aging effects on tissues including the arteries, heart, and kidneys. The subclinical inflammatory process affecting patients with atherosclerotic disease is thought to be responsible for the decreased expression and activation of Klotho, suggesting that this biomarker could be a potential target to counteract the damaging effects of inflammation on tissues [226,227].

In conclusion, the increasing prevalence of lower-extremity PAD with its inevitable consequences on morbidity, mortality, and healthcare costs requires novel effective therapies to control this disabling disease. Despite evidence-based interventions, harmful habits correction, and best medical care, some patients still experience high rates of MACEs and MALEs due to their residual risk not being covered by current recommended treatments.

Immunomodulatory agents directed at PAD are not available yet. However, understanding the anti-inflammatory effects of the current available therapies could suggest new strategies to improve the management of patients by broadening the treatment horizon.

## 13. Conclusions

In view of the need for new therapeutic strategies to prevent and slow the pandemic spread of PAD, it is crucial to understand the pleiotropic effects of the currently available treatments (Figure 1). Inflammation contributes substantially to the residual risk of PAD patients and all efforts should be directed towards the development of new potential therapies targeting the mechanisms responsible for disease progression despite the current optimal therapy.

## Figures and Tables

**Figure 1 ijms-24-16099-f001:**
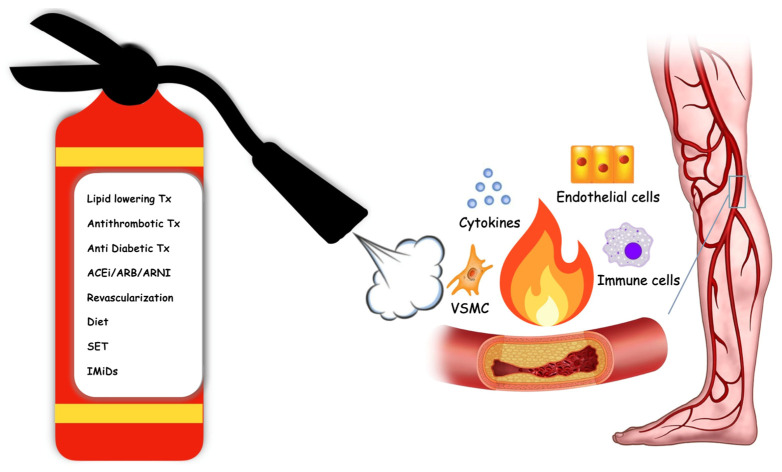
Schematic representation of the impact of current medical treatment and subclinical chronic inflammation on the development and progression of peripheral artery disease in the lower limbs. Tx: therapy, ACEi: ACE inhibitors, ARB: angiotensin II receptor blockers, ARNI: angiotensin receptor–neprilysin inhibitor, SET: supervised exercise therapy, IMiDs: immunomodulatory therapy.

**Table 1 ijms-24-16099-t001:** This table provides an overview of various medical treatments used in the management of peripheral artery disease and their respective impacts on subclinical chronic inflammation. The impact of each treatment may vary, and the table provides a broad summary of their effects.

Table	Downregulated Pathways	Upregulated Pathways	Currently Measured Clinical Outcomes in PAD	References
Statins	Attraction of inflammatory cells and activation of T-lymphocytes (by LFA-1), PAI-1 expression; Inhibitory effect on Rho and Ras GTP-ase	NO (by the inhibition of isoprenoids pathway), Th2 cell differentiation.	Increase in amputation-free survival; MACE reduction; Slower decline in walking capacity.	[36,37,38,39,40,41]
Ezetimibe	LDL absorption, TNF-α and IL-1β; Phosphorylated glycogen synthase kinase and Akt pathways.	-	Decreased cardiovascular risk in patients with polyvascular diseases, in particular if associated with statins.	[42,43,44,45]
PCSK9 inhibitors	Foam cell formation, cytokine release, monocyte recruitment, and TLR4/NF-κB pathway; Inhibition of the ATP-binding cassette transporter; LDL-c, PCSK9 function.	-	Reduced MACEs; Reduced MALEs.	[46,47,48]
Cilostazol and Prostaglandin	TNF-α, VCAM-1, MMP-9, PAI-1, hsCRP, and adiponectin; Thromboxane synthesis.	eNOS.	Improvement in walking distance in patients with intermittent claudication.	[49,50,51,52]
Aspirin	Thromboxane, prostanoids, and PGE2.	Lipoxins and PGI2.	MACE reduction with only minimal benefit on MALE prevention.	[53,54]
Clopidogrel	CRP, IL-6, TNF-α, p-selectin, CD63, and RANTES (CCL5).	-	MACE reduction.	[55,56,57]
Low-dose Rivaroxaban	PARs activation, superoxide anion, and NF-κB.	eNOS.	MACE and MALE reduction; Reduction in thrombo-inflammation, even in distal lesions unsuitable for invasive revascularization.	[58,59,60]
SGLT2-i	NF-κB, ROS, pro-inflammatory adipocytokines, IL-6, and uric acid.	eNOS and macrophages M2.	Lower MACEs with no increase in MALE incidence.	[61,62,63,64,65,66]
GLP-1a	Plasminogen activator inhibitor, LDL, triglycerides and hsCRP, and MMPs.	SIRT6 expression and collagen levels.	Increase in percentage flow mediated diameter.	[33,67]
ACEi/ARBs/ARNI	ROS, NF-κB, chemokines, and adhesion molecules.	eNOS.	Improvement in maximum walking distance and pain-free walking with a delay in PAD progression.	[61,68,69]
Diet				[35,70,71,72,73,74,75,76,77,78,79]
Plant-based diets	TNF-α, IL-6, IL-1β, and chemokines.	-	-	
Micronutrients	Risk of rupture of vulnerable plaque, proliferation of VSMCs.	-	MACE and MALE reduction.	
High intake of protein diet	-	LDL-c levels increase.	-	
PUFAs	LDL-c oxidative stress, NF-κB, V-CAM-1, TMAO-mediated vascular inflammation.	Soluble fiber intake: short-chain fatty acids and modulation of PPAR-α.	-	
Supervised exercise therapy (SET)	Insulin resistance, oxidative stress, E -selectin, and ICAM-I.	Improvement in lipid profile and glucose metabolism.	Remarkable relief form claudication with an increase in walking performance.	[80,81]
Immunomodulatory therapy				(ClinicalTrials.gov identifier: NCT04774159).
Low dose colchicine	NLRP3 inflammasome; IL-1β and IL-18.	-	Under investigation.	

Abbreviations: NO, Nitric oxide; MACE, Major adverse cardiovascular events; LFA-1, Lymphocyte function-associated antigen-1; PAI-1, plasminogen activator inhibitor-1; GTP, Guanosine triphosphate; LDL, Low-density lipoprotein; TNF, Tumor necrosis factor; IL, Interleukin; TLR, Toll-like receptor; NF-κB, Nuclear factor kappa-light-chain-enhancer of activated B cells; ATP, Adenosine triphosphate; PCSK9, proprotein convertase subtilisin/kexin type 9; VCAM-1, Vascular cell adhesion protein 1; MMP, Matrix Metalloproteinase; CRP, C-reactive protein; eNOS, Endothelial nitric oxide synthase; PGE2, Prostaglandin E2; RANTES, Regulated upon activation, normal T cell expressed and secreted; CCL5, CC-chemokine ligand 5; PARs, Protease-activated receptor; MALE, Major adverse limb events; ROS, Reactive oxygen species; SIRT, Sirtuin; VSMCs, Vascular smooth muscle cells; TMAO, Trimethylamine N-oxide; PPAR, Peroxisome proliferator-activated receptors; ICAM-1, Intercellular adhesion molecule-1; NLRP3, Nucleotide-binding domain, leucine-rich–containing family, pyrin domain–containing-3.

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
