# Peer review of "Current Medical Therapy and Revascularization in Peripheral Artery Disease of the Lower Limbs: Impacts on Subclinical Chronic Inflammation"

_ijms, 2023, doi:10.3390/ijms242216099_

Round 1
Reviewer 1 Report
Comments and Suggestions for Authors
This is a review of potential molecular mechanisms of inflammation that might be affected by current interventions for PAD. It is thoughtful and comprehensive; however, I have some minor suggestions.
In the text there are multiple instances where the table is referenced for providing some data on current outcomes associated with specific interventions.
Examples:
SGLT2-I and GLP-1 promote NO production and subsequent vasodilation (Table)
Patients on ACEi had significant improvement in walking distance (Table)
The effects of therapies on these outcomes are not shown in the table. It would be helpful to include a column in the table that indicates effects of these treatments on currently measured clinical outcomes in PAD patients, i.e., “improvement of flow-mediated dilation” or “improvement of maximum walking distance and pain free walking distance”
Line 234 “percentage flow mediated diameter” should be replaced with “normalized flow-mediated dilation”
Line 441 How is PAD included as a cerebrovascular disease. Should CVDs be an abbreviation for cardiovascular disease?
Some further editing is needed throughout the text for minor grammatical errors.
Comments on the Quality of English LanguagePlease check carefully for some minor grammatical errors throughout the manuscript.
Author Response
We thank the Editor for her/his assessment. We are pleased that our manuscript was rated as potentially acceptable for publication in the International Journal of Molecular Sciences, subject to adequate revision and response to the comments raised by the Reviewers.
We have revised the manuscript based on the comments made by the Reviewers, making an effort to address all queries. Please also find our point-by-point response to the comments raised by the Reviewers.
We would like to take this opportunity to express our sincere thanks to the Reviewers for identifying areas of our manuscript needing modification. We would also like to thank you for allowing us to resubmit a revised copy of the manuscript. We hope that the revised manuscript is acceptable for publication in the International Journal of Molecular Sciences.
_____________________
Reviewer 1
This is a review of potential molecular mechanisms of inflammation that might be affected by current interventions for PAD. It is thoughtful and comprehensive; however, I have some minor suggestions.
Authors’ response
We thank the reviewer for their significant contribution to the improvement of our work. Please find more details below on how we approached each point brought up by the reviewer
- In the text there are multiple instances where the table is referenced for providing some data on current outcomes associated with specific interventions.
Examples:
- SGLT2-I and GLP-1 promote NO production and subsequent vasodilation (Table)
- Patients on ACEi had significant improvement in walking distance (Table)
The effects of therapies on these outcomes are not shown in the table. It would be helpful to include a column in the table that indicates effects of these treatments on currently measured clinical outcomes in PAD patients, i.e., “improvement of flow-mediated dilation” or “improvement of maximum walking distance and pain free walking distance”
Authors’ response
We thank the reviewer for this constructive comment. We added a column labeled “Currently measured clinical outcomes in PAD”. References in the were adjusted accordingly.
As an example:
|
Table |
Downregulated pathways
|
Upregulated pathways |
Currently measured clinical outcomes in PAD |
References |
|
Statins |
-Attraction of inflammatory cells and activation of T-lymphocytes (by LFA-1), PAI-1 expression. -Inhibitory effect on Rho and Ras GTP-asa. |
-NO (by the inhibition of isoprenoids pathway), Th2 cells differentiation. |
- Increase in Amputation free survival
- MACE Reduction
- slower decline in walking capacity |
(38, 39, 43-45, 47) |
- Line 234 “percentage flow mediated diameter” should be replaced with “normalized flow-mediated dilation”
Authors’ response
We appreciate the comment. We replaced the term. Please see Line 504 “Moreover, a significant increase in normalized flow-mediated dilation was seen in the initial phase of therapy”
- Line 442 How is PAD included as a cerebrovascular disease. Should CVDs be an abbreviation for cardiovascular disease?
Authors’ response
We thank the reviewer for the comment. By CVD we inclusively mean both cardiovascular and cerebrovascular diseases. For this reason, we added “cardiovascular disease” to “cerebrovascular disease”. Please see lines 631-632 “Diet remains the most cost-effective and successful way to prevent and treat CVDs (cerebrovascular and cardiovascular disease), including PAD”
Reviewer 2 Report
Comments and Suggestions for Authors
Summary- The paper aimed to review the literature for the impact of PAD treatments on inflammation. This is as residual inflammatory risk is considered to be an important reason behind the continued decline and death in people with all types of cardiovascular disease. Overall, the authors covered the topic comprehensively and I think it would be of interest to researchers and clincians in this area. There were some sections which felt disordered Some edits to order and content of the article would strengthen it.
Concerns
1) The abstract had a methods section stating "a comprehensive review of 218 original articles,". While I agree that the cover is comprehensive, this wording makes it sound like a systematic review, and I was surpsied to not see a methods section in the article. I suggest amending the abstract subheadings or content to prevent other readers forming the same assumption. Alternatively, a section on the methods could be added if the approach was systematic.
2) The table
i. placement was before the section on inflammation, or introducing any of the treatments. I recommend it be moved down to after line 166.
ii. The table also felt disordered as different treatments in the same broad category were not next to one another (e.g. statins, PCSK1, Ezetimibe). They also didn't align with the order of the headings in the text. Changing table order to match text, or be grouped by similar categories would help.
iii. The table would be more informative with a little more detail on what inflammatory markers or pathways went up/down, rather than listing them without this direction.
iv. Lastly, depending on table final placement, abbreviations used in table may need to be defined below.
3) The discussion introduced new information, including a large section on antiinflammatory medications. The discussion should not introduce new information, but bring together the meaning (and limitations) of the sections before. I recommend
i. Moving information from here to the relevant sections in aritcle body (e.g. statins affecing hsCRP belongs in statins section)
ii. Making a section on anti-inflammatory targeting,
iii. and then revising the dicussion to ensure the focus is on understanding the information presented, not adding new information.
Recommendations- some minor amendments
line 88/89 “Along with lifestyle correction, pharmacological therapy is a mainstay for PAD management. low density lipoprotein cholesterol (LDL-c). More recently” I think the inclusion of "low density lipiprotein" in the middle here is an error
line 89-91 "symptomatic patients with PAD, antithrombotic therapy may further reduce the mortality and disabling consequences resulting from cardiovascular accidents(17)" What are cardiovascular accidents? Maybe "cardiovascular events"?
Lines 92-95 add reference
Line 96 “CLTI often leads to loss of limbs, death, and significantly lower quality of life.” Probably rearrange to put "death" last so it goes in order of severity?
Line 104- Typo in SET as "Supervisted physiotherapy" instead of "supervised exercise"
Line 147- define "MACE"
Line 491 "Emotional eating (binge eating, anorexia nervosa)(126, 127)," I have concerns about anorexia being listed as "emotional eating", it is an eating disorder and mental health problem. I think the bracketed section would be best removed here.
Line 725 "LEAD" is used. Is this distinct from PAD? I understand peripheral is not strictly "lower" but it seemed unusual to introduce a different abbreviation for a similar concept.
Comments on the Quality of English LanguageThe English was easy to comprehend throughout. There were some minor errors throughout, a few examples given below. Whilenone of these affected comprehension, I recommend the authors review to locate and correct any errors.
Line 46 “This residual risk and disease pro-gression suggest a plausible is resistance to traditional therapies that failed to address other pathophysiological disease pathways (6).” remove "is"
Line 50/51 “not only starts the development of atherosclerotic plaque but also accelerate its progression(7)” should be “accelerates”
line 63- “Heart disease understanding and perception by the general population is increasing to by improvement social awareness.” “increasing too”?
line 64" On the contrary, PAD of lower limbs remains mostly underdiagnosed, and poorly understood, and an often neglected medical condition by(11)." remove "by"
line 67 “It is estimated that so over 230 million people” remove “so”
line 106 "
106- “improved of quality of life” remove “of"
Author Response
We thank the Editor for her/his assessment. We are pleased that our manuscript was rated as potentially acceptable for publication in the International Journal of Molecular Sciences, subject to adequate revision and response to the comments raised by the Reviewers.
We have revised the manuscript based on the comments made by the Reviewers, making an effort to address all queries. Please also find our point-by-point response to the comments raised by the Reviewers.
We would like to take this opportunity to express our sincere thanks to the Reviewers for identifying areas of our manuscript needing modification. We would also like to thank you for allowing us to resubmit a revised copy of the manuscript. We hope that the revised manuscript is acceptable for publication in the International Journal of Molecular Sciences.
_____________________
Reviewer 2
The paper aimed to review the literature for the impact of PAD treatments on inflammation. This is as residual inflammatory risk is considered to be an important reason behind the continued decline and death in people with all types of cardiovascular disease. Overall, the authors covered the topic comprehensively and I think it would be of interest to researchers and clinicians in this area. There were some sections which felt disordered. Some edits to order and content of the article would strengthen it.
Author’s response
We thank the reviewer for the thoughtful comment. We restructured the paragraphs of the main body following a more logical approach, categorizing the different classes of intervention by their major mechanism of action or use in specific disorders. The table was then restructured accordingly, following the same order of the paragraphs in the main body.
Concerns
- The abstract had a methods section stating "a comprehensive review of 218 original articles,". While I agree that the cover is comprehensive, this wording makes it sound like a systematic review, and I was surprised to not see a methods section in the article. I suggest amending the abstract subheadings or content to prevent other readers forming the same assumption. Alternatively, a section on the methods could be added if the approach was systematic.
Author’s response
We thank the reviewer for the relevant comment. We deleted the number of articles reviewed, as our approach did not include any analysis so to consider the paper a systematic review.
We also removed the subheadings, per associate editor’s comment sent via e-mail.
- The table
Authors’ response
We really appreciate this comment, and we significantly improved the layout and location of our table. See Table 1 after line 758 for reference. Here is one row for a quick look at the improvements done.
|
Table |
Downregulated pathways
|
Upregulated pathways |
Currently measured clinical outcomes in PAD |
References |
|
Statins |
-Attraction of inflammatory cells and activation of T-lymphocytes (by LFA-1), PAI-1 expression. -Inhibitory effect on Rho and Ras GTP-asa. |
-NO (by the inhibition of isoprenoids pathway), Th2 cells differentiation. |
- Increase in Amputation free survival
- MACE Reduction
- slower decline in walking capacity |
(38, 39, 43-45, 47) |
Please see more details in the comments below:
- placement was before the section on inflammation, or introducing any of the treatments. I recommend it be moved down to after line 166.
Authors’ response: We moved the table at the end of the main body, so to provide a final summary of what stated so far.
- The table also felt disordered as different treatments in the same broad category were not next to one another (e.g. statins, PCSK1, Ezetimibe). They also didn't align with the order of the headings in the text. Changing table order to match text, or be grouped by similar categories would help.
Authors’ response: We initially restructured the main body, creating macro areas regarding the major mechanisms of action of the interventions (ex. Lipid lowering therapy, antithrombotic therapy, etc.).
We then reorganized the table according to the reviewer suggestion and the new order of the paragraphs.
- The table would be more informative with a little more detail on what inflammatory markers or pathways went up/down, rather than listing them without this direction.
Authors’ response: We divided the markers in two different columns, named “Downregulated pathways” and “Upregulated pathways”, so to better distinguish the effect of each single intervention.
- Lastly, depending on table final placement, abbreviations used in table may need to be defined below.
Authors’ response: Given the table has been placed after the main body, the issues with abbreviation should have been solved.
- The discussion introduced new information, including a large section on antiinflammatory medications. The discussion should not introduce new information, but bring together the meaning (and limitations) of the sections before.
Author’s response
We thank the reviewer for their comments. Please find more details below:
- Moving information from here to the relevant sections in article body (e.g. statins affecting hsCRP belongs in statins section)
Authors’ response: The actions of different interventions on molecular pathways were already included in the main body.
- Making a section on anti-inflammatory targeting
Authors’ response: By redesigning the overall structure of the main body, we added a new section named “ 8. IMMUNOMODULATORY THERAPY: promising perspectives”. Please see lines 698-756
- and then revising the dicussion to ensure the focus is on understanding the information presented, not adding new information.
Authors’ response: We gave the discussion a more structured organization, introducing the following subheadings
- The Residual Cardiovascular Risk
- Current available GDMT to manage the residual inflammatory risk
- Future perspectives to manage the residual inflammatory risk
- line 88/89 “Along with lifestyle correction, pharmacological therapy is a mainstay for PAD management. low density lipoprotein cholesterol (LDL-c). More recently” I think the inclusion of "low density lipiprotein" in the middle here is an error
Authors’ Response
We thank the reviewer for the comment. We deleted the “low density lipoprotein”
- line 89-91 "symptomatic patients with PAD, antithrombotic therapy may further reduce the mortality and disabling consequences resulting from cardiovascular accidents(17)" What are cardiovascular accidents? Maybe "cardiovascular events"?
Authors’ Response
We thank the reviewer for the comment. We changed “Cardiovascular accidents” with “cardiovascular events”
- Lines 92-95 add reference
Authors’ Response
We thank the reviewer for the comment.We added the following reference:
- Aboyans V, Chastaingt L. What LEADs to the under-treatment of patients with lower-extremity artery disease? Eur J Prev Cardiol. 2023;30(11):1090-1.
- Line 96 “CLTI often leads to loss of limbs, death, and significantly lower quality of life.” Probably rearrange to put "death" last so it goes in order of severity?
Authors’ Response
We thank the reviewer for the comment. We rephrased as below:
CLTI often leads to loss of limbs, significantly lower quality of life and even death.
- Line 104- Typo in SET as "Supervisted physiotherapy" instead of "supervised exercise"
Authors’ Response
We thank the reviewer for the comment. We changed "Supervised physiotherapy" to "supervised exercise"
- Line 147- define "MACE"
Authors’ Response
We thank the reviewer for the comment. We added “Major Adverse Cardiac Event (MACE)”
- Line 491 "Emotional eating (binge eating, anorexia nervosa)(126, 127)," I have concerns about anorexia being listed as "emotional eating", it is an eating disorder and mental health problem. I think the bracketed section would be best removed here.
Authors’ Response
We thank the reviewer for the comment. We removed the bracketed sections.
- Line 725 "LEAD" is used. Is this distinct from PAD? I understand peripheral is not strictly "lower" but it seemed unusual to introduce a different abbreviation for a similar concept.
Authors’ Response
We thank the reviewer for the comment. We changed the term LEAD with PAD to be more consistent with the rest of the paPER
Comments on the Quality of English Language
The English was easy to comprehend throughout. There were some minor errors throughout, a few examples given below. While none of these affected comprehension, I recommend the authors review to locate and correct any errors.
Authors’ Response
We appreciate the comments and we corrected several errors throughout the text. Please find more details below.
- Line 46 “This residual risk and disease pro-gression suggest a plausible is resistance to traditional therapies that failed to address other pathophysiological disease pathways (6).” remove "is"
Authors’ Response
We removed “is”
- Line 50/51 “not only starts the development of atherosclerotic plaque but also accelerate its progression(7)” should be “accelerates”
Authors’ Response
We substituted “accelerate” with “accelerates”
- line 63- “Heart disease understanding and perception by the general population is increasing to by improvement social awareness.” “increasing too”?
Authors’ Response
We rephrased as following (Lines 63-64). “Heart disease understanding and perception by the general population is increasing by improvement of social awareness.”
- line 64" On the contrary, PAD of lower limbs remains mostly underdiagnosed, and poorly understood, and an often neglected medical condition by(11)." remove "by"
Authors’ Response
We removed “by”
- line 67 “It is estimated that so over 230 million people” remove “so”
Authors’ Response
We removed “so”
- 106- “improved of quality of life” remove “of"
Authors’ Response
We removed “so”
Reviewer 3 Report
Comments and Suggestions for Authors
This manuscript is a review of PAD and current therapies and the potential role played by inflammation in this disease. Comments are outlined below:
Line 47 - "suggest...resistance" something is missing in this sentence - please revise to clarify
lines 63-64 - is increasing...awareness - something missing here - please revise to clarify
line 89 - low density...(LDL-c) is out of place, or something is missing.
Table - I found the table a bit confusing - It would help if you indicated in the table which effects were inhibitory and which effects were stimulatory to the processes listed.
line 203 - "proved" probably should be repaced with "shown".
lines 437-439 - Please cite a reference for this last statement.
line 442 - Should this be limited to cerebrovascular disease or extended to all vascular diseases? Please clarify?
line 459 - "exert contribute"?
lines 535-537 - You might note that it is not surprising that iloprost has a similar action as cilostazol because isolprost acts through the IP receptor which is a Gs-PCR that is coupled to adenylate cyclase and production of cAMP.
Lines 552-570 - Given the positive effects of liloprost in PAD, it is a bit counterintutive that aspirin which inhibits prostacyclin production would also have positive effects - this should be discussed in a sentence or 2.
Discussion - I found the discussion to be somewhat confusing. Perhaps subheadings could be used to better organize this section and the section could be revised to increase clarity and focus?
Comments on the Quality of English LanguageThere were a number of typographical errors in the manuscript (see comments for a few examples). These and non-standard English in some instances made sections of the manuscript difficult to understand.
Author Response
We thank the Editor for her/his assessment. We are pleased that our manuscript was rated as potentially acceptable for publication in the International Journal of Molecular Sciences, subject to adequate revision and response to the comments raised by the Reviewers.
We have revised the manuscript based on the comments made by the Reviewers, making an effort to address all queries. Please also find our point-by-point response to the comments raised by the Reviewers.
We would like to take this opportunity to express our sincere thanks to the Reviewers for identifying areas of our manuscript needing modification. We would also like to thank you for allowing us to resubmit a revised copy of the manuscript. We hope that the revised manuscript is acceptable for publication in the International Journal of Molecular Sciences.
_____________________
Reviewer 3
This manuscript is a review of PAD and current therapies and the potential role played by inflammation in this disease. Comments are outlined below
Authors’ Response
We thank the reviewer for their feedback. Please find more detail below
- Line 47 - "suggest...resistance" something is missing in this sentence - please revise to clarify
Authors’ Response
We removed “is” so that the new sentence is as following: This residual risk and disease progression suggest a plausible resistance to traditional therapies that failed to address other pathophysiological disease pathways (6).
- lines 63-64 - is increasing...awareness - something missing here - please revise to clarify
Authors’ Response
We rephrased as following “Heart disease understanding and perception by the general population is increasing by improvement of social awareness”
- line 89 - low density...(LDL-c) is out of place, or something is missing.
Authors’ Response
We removed “low density lipoprotein cholesterol (LDL-c)”
- Table - I found the table a bit confusing - It would help if you indicated in the table which effects were inhibitory and which effects were stimulatory to the processes listed.
Authors’ response: We divided the markers in two different columns, named “Downregulated pathways” and “Upregulated pathways”, so to better distinguish the effect of each single intervention. Please see Table 1 after line 758. Here is just one row to provide an example.
|
Table |
Downregulated pathways
|
Upregulated pathways |
Currently measured clinical outcomes in PAD |
References |
|
Statins |
-Attraction of inflammatory cells and activation of T-lymphocytes (by LFA-1), PAI-1 expression. -Inhibitory effect on Rho and Ras GTP-asa. |
-NO (by the inhibition of isoprenoids pathway), Th2 cells differentiation. |
- Increase in Amputation free survival
- MACE Reduction
- slower decline in walking capacity |
(38, 39, 43-45, 47) |
- line 203 - "proved" probably should be repaced with "shown".
Authors’ Response
We replaced “proved” with shown
- lines 437-439 - Please cite a reference for this last statement.
Authors’ Response
We thank the reviewer for the comment. We added the following reference:
- Issa N. Making a Case for the Anti-inflammatory Effects of ACE Inhibitors and Angiotensin II Receptor Blockers: Evidence From Randomized Controlled Trials. Mayo Clin Proc. 2022;97(10):1766-8.
- line 442 - Should this be limited to cerebrovascular disease or extended to all vascular diseases? Please clarify?
Authors’ Response
We rephrased accordingly. Please see lines 621-622
Diet remains the most cost-effective and successful way to prevent and treat CVDs (cerebrovascular and cardiovascular disease), including PAD.
- line 459 - "exert contribute"?
Authors’ Response
We removed “contribute”. Please see lines 643-644. ”The healing effects on ischemic ulcer and immune modulation during ischemic triggers of vitamin C and D supplementation respectively contribute to the reduction of MACE and MALE(161, 162)”
- lines 535-537 - You might note that it is not surprising that iloprost has a similar action as cilostazol because isolprost acts through the IP receptor which is a Gs-PCR that is coupled to adenylate cyclase and production of cAMP.
Authors’ Response
We included the reviewer comment by further elaborating on the similarities. Please see lines 313-319.
“As noted by Lessiani et al., Iloprost, a structural analog of prostacyclin, has known anti-inflammatory, antithrombotic, and vasodynamic effects that can be similarly mimicked by cilostazol. In fact, the inhibition of platelet function and the improvement of endothelial cell functions is mediated by a selective inhibition of the phosphodiesterase type 3 molecular pathway with consequent increase in cAMP production. Indeed, a decreased level of soluble CD40L was observed accompanied by increased plasma nitrates. Moreover, iloprost treatment reduced residual thromboxane synthesis(81)”
- Lines 552-570 - Given the positive effects of liloprost in PAD, it is a bit counterintutive that aspirin which inhibits prostacyclin production would also have positive effects - this should be discussed in a sentence or 2.
Authors’ Response
We included the reviewer observation as following in lines 331-340 of the manuscript.
“Aspirin causes irreversible inactivation of cyclooxygenase (COX) enzyme types 1 and 2 by acetylating its serine residue. Thus, aspirin causes a reduction in the synthesis of prostaglandins and thromboxane. The main effect is a significant reduction of proinflammatory and prothrombotic effects mediated by some isotypes of prostanoids. Furthermore, COX2 acetylated by aspirin produces lipoxins that foster an anti-inflammatory effect. The antithrombotic effects of low doses of acetylsalicylic acid are mediated by the ability to largely reduce the formation of thromboxane A2 in platelets, while the synthesis of anti-inflammatory prostanoids (e.g., prostaglandin I2) is only minimally affected. Additionally, aspirin-modified COX-2 contributes to further increase the expression of lipoxins and prostacyclins with a net anti-inflammatory and anti-thrombotic effect (85).”
- Discussion - I found the discussion to be somewhat confusing. Perhaps subheadings could be used to better organize this section and the section could be revised to increase clarity and focus?
Authors’ Response
We gave the discussion a more structured organization, introducing the following subheadings
- The Residual Cardiovascular Risk
- Current available GDMT to manage the residual inflammatory risk
- Future perspectives to manage the residual inflammatory risk
- Comments on the Quality of English Language
There were a number of typographical errors in the manuscript (see comments for a few examples). These and non-standard English in some instances made sections of the manuscript difficult to understand.
Authors’ Response
We thank you the reviewer for the important suggestion. We did a thorough revision of the manuscript and corrected several English errors. We also tired to review the syntax of certain paragraphs that may have been harder to read.